# Decomposing Densification in Gaussian Splatting for Faster 3D Scene Reconstruction

## Abstract

3D Gaussian Splatting (GS) has emerged as a powerful representation for high-quality scene reconstruction, offering compelling rendering quality. However, the training process of GS often suffers from slow convergence due to inefficient densification and suboptimal spatial distribution of Gaussian primitives. In this work, we present a comprehensive analysis of the split and clone operations during the densification phase, revealing their distinct roles in balancing detail preservation and computational efficiency. Building upon this analysis, we propose a global-to-local densification strategy, which facilitates more efficient growth of Gaussians across the scene space, promoting both global coverage and local refinement. To cooperate with the proposed densification strategy and promote sufficient diffusion of Gaussian primitives in space, we introduce an energy-guided coarse-to-fine multi-resolution training framework, which gradually increases resolution based on energy density in 2D images. Additionally, we dynamically prune unnecessary Gaussian primitives to speed up the training. Extensive experiments on MipNeRF-360, Deep Blending, and Tanks & Temples datasets demonstrate that our approach significantly accelerates training—achieving over 2x speedup with fewer Gaussian primitives and superior reconstruction performance.

## 1 Introduction

Reconstructing high-quality 3D representations from unordered image collections remains a fundamental challenge in computer vision and graphics. Neural radiance fields (NeRF) Mildenhall et al. (2020) have revolutionized this domain through their implicit scene representation paradigm, combining deep learning with volumetric rendering to achieve unprecedented view synthesis quality Oechsle et al. (2021); Park et al. (2021); Wang et al. (2021); Yariv et al. (2021). Despite these successes, the computational demands of ray-based volumetric rendering present critical limitations. The requirement for dense spatial sampling along viewing rays significantly hinders both training convergence and rendering efficiency. Recent advancements in 3D reconstruction have highlighted 3D Gaussian Splatting (3DGS) Kerbl et al. (2023) as a promising technique for high-fidelity scene modeling. By representing scenes as collections of anisotropic Gaussian primitives, plenty of works Waczynska et al. (2024); Yan et al. (2024); Yang et al. (2024) based on the 3DGS achieve impressive visual quality with explicit geometric distribution and efficient rendering pipelines. However, there still exists an urgent requirement for computational efficiency improvements to deploy 3D Gaussian Splatting on resource-constrained devices or enable its practical application in real-time reconstruction and dynamic modeling scenarios where training time constitutes a critical bottleneck Cong et al. (2025); Javed et al. (2024); Tan et al. (2024).

Based on the 3DGS pipeline, several recent approaches Chen et al. (2025); Fang & Wang (2024a); Hanson et al. (2024); Mallick et al. (2024) have pursued optimization acceleration from the perspectives of geometric distribution, optimizer, and multi-resolution and so on. Taming 3DGS Mallick et al. (2024) makes each tile uses a parallelization scheme over the 2D splats instead of pixels. Mini-splatting Fang & Wang (2024a) utilizes depth to achieve efficient reinitialization of the Gaussian primitives from the perspective of spatial geometry. EDC Deng et al. (2024) proposes a long-axis split operation and a pruning strategy to efficiently control the Gaussian densification. DashGaussian Chen et al. (2025) introduce a resolution scheduler and a primitive scheduler to accelerate the training time.

In this work, we systematically analyze the bottlenecks in 3D Gaussian Splatting reconstruction, particularly focusing on inefficient spatial spread and redundant Gaussian primitives during optimization. We reveal that the split operation takes charge of the global spread while the clone operation governs the local refinement (cf. Table 1 and Fig. 3). Then We identify clone operations in the early stage as the primary cause of excessive Gaussian clustering during optimization, where redundant primitives aggregate while contributing minimally to reconstruction fidelity (cf. Fig. 2). To address these limitations, we propose a global-to-local strategy that decouples split and clone across densification phases.

We further design a energy-aware multi-resolution training strategy to facilitate this global-to-local optimization strategy. Specifically, we promote Gaussian primitives' global spread with split operation at the lower resolution and suppress clone operations. This prevents premature local clustering and ensures efficient scene coverage. Only when transitioning to full-resolution training do we reintroduce clone operations to refine high-frequency details. Additionally, we integrates an opacity pruning strategy with a adaptive threshold to remove the unnecessary Gaussian primitives. The pipeline is shown in Fig. 1. By jointly utilizing the proposed approaches, our method achieves an approximately $2\times$ acceleration in training speed compared to baseline implementations, with a comparable or even better reconstruction quality. In summary, our contributions are as follows:

- We first reveal the split takes charge of the global spread and the clone governs the local refinement and propose a global-to-fine densification to accelerate the optimization.
- We introduce a energy-aware multi-resolution framework to promote the global-to-fine densification for further acceleration.
- Comprehensive experiments conducted on three datasets demonstrate that our method achieves an approximately $2\times$ speedup over the baseline, while maintaining or even enhancing the performance.

## 2 RELATED WORKS

### 2.1 3D GAUSSIAN SPLATTING

3D Gaussian Splatting Kerbl et al. (2023) has emerged as a compelling approach for 3D scene reconstruction, enabling real-time rendering while preserving photorealistic quality. Unlike implicit neural fields (e.g., NeRF Mildenhall et al. (2020)) that rely on computationally intensive ray marching for volume rendering, 3DGS formulates scenes as collections of anisotropic Gaussian primitives with full covariance matrices. This explicit representation allows efficient tile-based rasterization through differentiable projection and alpha blending, bypassing the limitations of neural rendering pipelines.

In recent years, there has been a surge in research efforts that have pushed forward the technological frontiers of 3DGS across multiple domains, with particularly transformative impacts on human avatar generation Cha et al. (2024); Jiang et al. (2024); Lyu et al. (2024); Zielonka et al. (2025), Autonomous Driving Chen et al. (2024); Hess et al. (2024); Lei et al. (2025); Zhou et al. (2024), and photorealistic scene renderings Chao et al. (2024); Cheng et al. (2024); Xie et al. (2024); Xu et al. (2024).

### 2.2 ACCELERATION FOR 3DGS OPTIMIZATION

Although the rendering speed of 3D Gaussian Splatting is much faster than that of NeRF, it still takes tens of minutes to complete the rendering of a scene on a high-performance GPU. Plenty of subsequent works accelerated the optimization process from the perspectives of optimization strategies, the number of Gaussian spheres, etc. Taming 3DGS Mallick et al. (2024) reformulates the original per-pixel parallelization into per-splat parallel backpropagation, significantly accelerating the optimization process of 3D Gaussian Splatting and establishing a strong baseline for following research. Mini-Splatting Fang & Wang (2024b) saves the training time and memory cost by maintaining the most important primitive for each pixel through depth reinitialization. Speedy-Splat Hanson et al. (2024) calculates a precise tile allocation of Gaussians when projected to the 2D image planes and prunes a fixed high proportion of Gaussians in specific iterations. Meanwhile,

reducing the resolution of renderings in the optimization stage is also a promising option for acceleration. EAGLES Girish et al. (2023) adopts several schedules for gradually increasing the resolution empirically. From the perspective of the frequency domain, DashGaussian Chen et al. (2025) designs a resolution scheduler and a primitive scheduler to efficiently reconstruct the scene from low frequency to high frequency.

However, these methods adopt the default densification strategy and do not explore the actual roles of split and clone. A similar work EDC Deng et al. (2024) replaced the clone operation with a proposed long-axis split based on AbsGS Ye et al. (2024) with limited improvement of training speed. In contrast, our methods analyze the behaviors of the split and clone operations and propose a global-to-local densification strategy to facilitates efficient growth of Gaussians across the scene. Then we design an energy-guided coarse-to-fine multi-resolution training framework to cooperate with the proposed densification strategy.

## 3 PRELIMINARY

**3D Gaussian Splatting**  3DGS Kerbl et al. (2023) represents 3D scenes through anisotropic Gaussian primitives and demonstrates state-of-the-art performance in both visual quality and rendering efficiency. Each Gaussian primitive $\mathcal{G}_i$ is formally defined by five core contributions: spatial position $\boldsymbol{u}_i \in \mathbb{R}^3$, opacity $\alpha_i$, orthogonal rotation matrix $\boldsymbol{R}_i \in \mathbb{R}^{3\times3}$, diagonal scaling matrix $\boldsymbol{S}_i \in \mathbb{R}^{3\times3}$, and spherical harmonics (SH) coefficients for view-dependent color representation. The Gaussian distribution is mathematically expressed as:

$$\mathcal{G}_i(\boldsymbol{x}) = \exp\left(-\frac{1}{2}(\boldsymbol{x} - \boldsymbol{u}_i)^T \Sigma_i^{-1}(\boldsymbol{x} - \boldsymbol{u}_i)\right), \tag{1}$$

where $\Sigma_i = \boldsymbol{R}_i \boldsymbol{S}_i \boldsymbol{S}_i^T \boldsymbol{R}_i^T$ ensures positive semi-definiteness. For real-time rendering, Gaussian primitives are projected onto the 2D image plane with the Jacobian affine approximation. Given camera extrinsic parameters $\boldsymbol{W}$ and projection matrix Jacobian $\boldsymbol{J}$, the 2D covariance in screen space becomes $\Sigma_i' = \boldsymbol{J}\boldsymbol{W}\Sigma_i\boldsymbol{W}^T\boldsymbol{J}^T$. The final pixel color $\mathcal{C}(\boldsymbol{p})$ is computed via alpha compositing of depth-sorted Gaussians:

$$\boldsymbol{C}(\boldsymbol{p}) = \sum_{i \in \mathcal{N}_{\boldsymbol{p}}} \boldsymbol{c}_i \sigma_i \prod_{j=1}^{i-1}(1 - \sigma_j), \quad \sigma_i = \alpha_i \mathcal{G}_i'(\boldsymbol{p}), \tag{2}$$

where $\boldsymbol{c}_i$ denotes SH-evaluated color and $\mathcal{N}_{\boldsymbol{p}}$ indexes visible Gaussians at pixel $\boldsymbol{p}$. The model is optimized using a hybrid loss combining $\mathcal{L}_1$ and structural similarity (i.e., D-SSIM term):

$$\mathcal{L}_{\text{total}} = (1 - \lambda)\|\boldsymbol{I} - \hat{\boldsymbol{I}}\|_1 + \lambda\mathcal{L}_{\text{D-SSIM}}(\boldsymbol{I}, \hat{\boldsymbol{I}}), \tag{3}$$

with default weight $\lambda = 0.2$, where $\boldsymbol{I}$ and $\hat{\boldsymbol{I}}$ denote ground truth and rendered images, respectively.

During the densification stage, the norm of the average position gradient of each Gaussian primitive is calculated every 100 iterations. If the gradient norm exceeds a predefined threshold, the corresponding Gaussian primitive will either be split or cloned. Specifically, if the maximum scale of the Gaussian exceeds a given scale threshold, it will be split into smaller components; otherwise, it is simply cloned with the same parameters.

## 4 METHODOLOGY

In this section, we present how the proposed method reduces optimization complexity to accelerate 3D Gaussian Splatting, while preserving rendering quality without compromise. In Sec. 4.1, we analyze the distinct behaviors of the split and clone operations and proposes a global-to-local densification strategy to improve optimization efficiency. In Sec. 4.3, we introduce a coarse-to-fine multi-resolution scheme based on the energy density in 2D images to better support the proposed densification strategy. In Sec. 4.4, we adopt an adaptive opacity threshold to better balance the trade-off between training efficiency and rendering quality.

### 4.1 GLOBAL-TO-LOCAL DENSIFICATION

To achieve the 3D reconstruction using the Gaussian Splatting, split and clone operations are applied simultaneously during the densification stage to densify and spread the Gaussian primitives spatially. We dig into the differences between the these two operations and claims two statements neglected by preview researches: 1. The split operation takes charge of the diffusion of the Gaussian primitives in space (cf. Sec. 4.1.1); 2. The number of Gaussian primitives produced by clone is much higher than that produced by split (cf. Sec. 4.1.2).

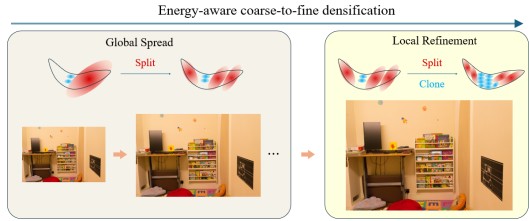

Figure 1: Pipeline of the global-to-local and coarse-to-fine densification.

### 4.1.1 SPATIAL DIFFUSION

In this section, we prove that the spatial diffusion is predominantly governed by the split operation, whereas the clone operation contributes to local feature refinement. We first revealed that the clone operation is the cause of the cluster phenomenon observed in Mini-Splatting Fang & Wang (2024a), which can be alleviated by our methods.

We regard the points extracted by structure from motion (SFM) for initialization as the parent points. Each newly generated Gaussian maintains a one-to-one correspondence with its parent point. During the optimization, We equip each Gaussian primitive with three more attributes: *original position*, *split count*, and *clone count*. The original position stores the initial coordinates of its parent point and the split/clone count quantify cumulative split/clone operations executed with respect to its parent point during optimization. After optimization, we classify Gaussians into three categories: split-dominated (split > clone), clone-dominated (clone > split), and equal (split = clone). Specifically, Gaussian $\mathcal{G}_A$ undergoes split to produce $\mathcal{G}_B$, and $\mathcal{G}_B$ clones to produce $\mathcal{G}_C$. Consequently, the parent point of $\mathcal{G}_C$ is $\mathcal{G}_A$ and $\mathcal{G}_C$ belongs to the equal category. We compute average Euclidean distances between final positions and initial positions of each gaussian for each category. As shown in Table 1, the average displacement distances across three datasets show that split-dominated Gaussians exhibit displacements around twenty times greater than clone-dominated countparts, demonstrating that spatial expansion is primarily driven by split operations. The camera extent is 1.1 times the radius of the smallest sphere covering all camera positions defined in the 3D Gaussian Splatting. It is regarded as a measurement representing the size of the scene. The displacement of clone-dominated primitives is less than 2% of the scene size, so we argue that the clone operation is mainly responsible for local refinement, while the split operation takes charge of the global diffusion.

Table 1: Average displacement distances for split-dominated, clone-dominated and equal primitives before and after optimization.

| Category | MipNeRF-360 Barron et al. (2022) | Deep Blending Hedman et al. (2018) | Tanks & Temples Knapitsch et al. (2017) |
|---|---|---|---|
| split-dominated | 2.42 | 0.75 | 2.40 |
| clone-dominated | 0.09 | 0.15 | 0.10 |
| equal | 0.17 | 0.23 | 0.15 |
| camera extent | 5.16 | 7.79 | 6.65 |

We present qualitative comparisons of Gaussian distributions across three densification strategies: (1) the standard adaptive approach from 3D Gaussian Splatting that dynamically selects splits/clones based on Gaussian scale, versus (2) split-only and (3) clone-only variants where all densification operations are enforced to use a single type. As shown in Fig. 2, clone-only version intensifies the local cluster phenomenon of the gaussian primitives (cf. bicycle frame and decoration on the wall) and fails to spread the Gaussian primitives, leading to a blurry rendering output due to insufficient spatial distribution (cf. houses in the distance of bicycle scene). Although the split-only variant produces a more uniform spatial distribution compared to other approaches, the complete lack of any clone operation prevents it from efficiently adapting to fine-grained scene details. As a result, it requires a significantly larger number of Gaussian primitives to adequately fit the scene, ultimately leading to a reconstruction quality that remains inferior to that of the baseline.

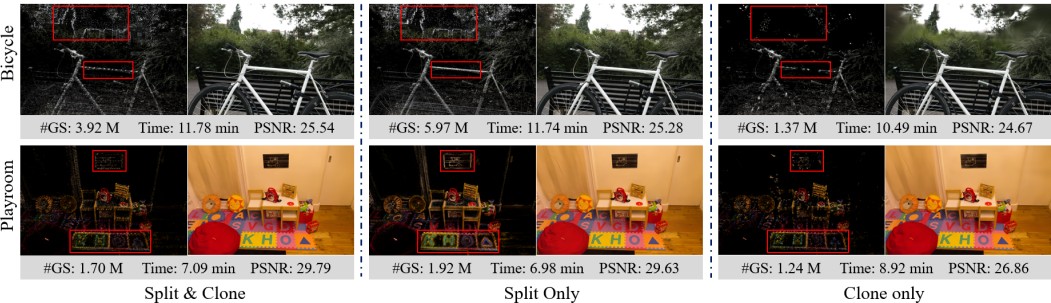

Figure 2: Visualization of the distribution of Gaussian primitives (left) and the rendered images (right) after optimization .

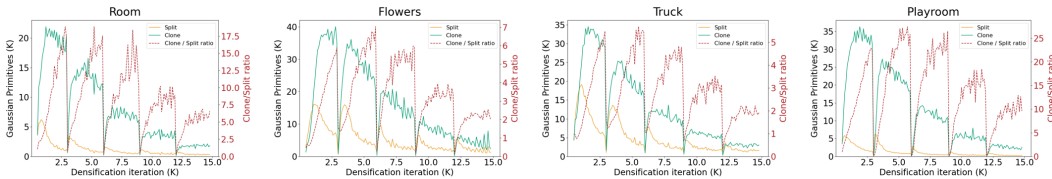

Figure 3: The number of Gaussian primitives generated through split and clone operations and the ratio of clone to split during the densification stage.

In summary, both quantitative results and qualitative visual analysis indicate that the split operation primarily takes charge of the global distribution of Gaussian primitives, while the clone operation is mainly responsible for the local refinement.

### 4.1.2 SPLIT-CLONE RATIO

We present quantitative analysis of Gaussian primitive evolution during the densification phase in 3D Gaussian Splatting. As demonstrated in Fig. 3, approximately $80\%$ of new primitives originate from cloning operations rather than splitting. To fit the rendering error caused by the opacity reset performed every 3k steps, both splits and clones were increased simultaneously. Subsequently, as the optimization progresses, the number of splits begins to decline, while the number of clones continues to rise. Comprehensive results across multiple scenes from different datasets confirm this trend.

According to the discover in the previous section, we argue that the local refinement at early densification phase cannot reduce the rendering error through clone operations, resulting in a persistently high gradient. As a result, a large number of Gaussian primitives are repeatedly cloned, causing computational redundancy and failing to improve the rendering quality.

These findings suggest two key implications: First, the majority of primitive growth stems from clone operations that may not contribute meaningfully to representation capacity in early stage. Second, the observed trend indicates potential for algorithmic optimization through adaptive densification strategies that can achieve computational efficiency with comparable performance.

### 4.2 GLOBAL-TO-LOCAL FRAMEWORK

Qualitative and quantitative experiments have proved that the clone operation used for local refinement has led to the generation of a large number of redundant Gaussian primitives, which is unnecessary for the optimization of the scene reconstruction in the early stage. Leveraging the empirically observed inverse correlation between Gaussian density and computational efficiency, We implement a two-phase densification framework: global spread and local refinement. At the first phase, we only apply the split operation to achieve fast and effective spatial distribution, leveraging

minimal Gaussian counts to minimize redundant computation on localized details. In the second phase, both the split and clone operations are employed. Given that a satisfactory spatial distribution has already been established, the clone operation enables efficient local refinement. The proposed phased training strategy can reduce training time by optimizing a substantially smaller number of primitives in the first phase.

To further enhance the effectiveness of the global-to-local densification process, we propose a coarse-to-fine multi-resolution scheduler based on the energy density in 2D images, which is elaborated in the following section. This approach eliminates the need for manually defining the boundary between the two phases.

### 4.3 COARSE-TO-FINE MULTI-RESOLUTION DENSIFICATION

The optimization of 3D Gaussian Splatting is conducted by projecting Gaussian primitives from 3D space to 2D pixel plane, where the rendering error is computed to update both their spatial distribution and attribute parameters. During the global spread phase of densification, we aim for a fast and efficient coverage of the target scene volume, without overemphasizing the reconstruction of fine image details at this early stage. Therefore, using full-resolution images for supervision can lead to unnecessary computational consumption and suboptimal behavior. Specifically, each pixel corresponds to a small 3D voxel, which encourages Gaussians primitives to converge prematurely to local optima by overfitting to individual pixels, thereby limiting their spatial expansion. Furthermore, when a large Gaussian projects to a large number of pixels, the accumulated gradient vectors may cancel each other out due to opposing vector directions in space, resulting in a small gradient error below the threshold and preventing further split operations Ye et al. (2024).

Inspired by frequency analysis techniques in 2D image processing and DashGaussian Chen et al. (2025), we propose a coarse-to-fine training strategy based on energy density to mitigate the aforementioned issues. Specifically, during the global spread phase, we train with downsampled images to enable efficient spatial diffusion of Gaussian primitives. Once a sufficient scene coverage is attained, we switch to full-resolution supervision for the local refinement phase, allowing accurate reconstruction of high-frequency image content.

We analyze the image energy distribution in frequency domain to design an adaptive resolution scheduling mechanism. Given an input image $\mathbf{I} \in \mathbb{R}^{H \times W \times C}$, we compute its energy spectrum through Fourier transform: $\mathcal{E}(\mathbf{I}) = \sqrt{\Re(\mathcal{F}(\mathbf{I}))^2 + \Im(\mathcal{F}(\mathbf{I}))^2}$, where $\mathcal{F}(\cdot), \Re, and \Im$ denotes 2D FFT, real part and imaginary part in frequency domain, and $\mathcal{E}(\cdot)$ calculates the energy density.

For multi-resolution analysis, we define a downscaling operator $\mathcal{D}_r(\cdot)$ that reduces spatial dimensions by factor $r$ using bilinear interpolation with anti-aliasing:

$$\mathbf{I}_r = \mathcal{D}_r(\mathbf{I}) = Bilinear(\mathbf{I}, \text{scale} = 1/r) \tag{4}$$

The energy density across resolutions is quantified as:

$$\mathcal{E}_r = \|\mathcal{E}(\mathbf{I}_r)\|_1 \cdot r^2 \tag{5}$$

where the scaling term $r^2$ normalizes energy values across different resolutions. Our resolution scheduler dynamically allocates training iterations based on energy ratios as follows:

$$\mathbf{T}_r = \text{Round}((\mathcal{E}_r/\mathcal{E}_1) \cdot \mathbf{T}_{\text{densify}}), r \in \mathcal{A} \tag{6}$$

where $\mathcal{A} = \{1, 2, ..., K\}$ denotes candidate scale factors, and $\mathbf{T}_r, \mathbf{T}_{\text{densify}}$ represents the allocated iteration at $r$-scaled resolution and total densification iteration. The training proceeds from coarsest $(r = K)$ to finest $(r = 1)$ resolution following reversed order. For scale factor $r$, the training starts at $\mathbf{T}_{r+1}$ and end at $\mathbf{T}_r$. This energy-aware strategy ensures optimal balance between global scene coverage at early phases and detail reconstruction in later phases.

### 4.4 ADAPTIVE OPACITY PRUNING

To prove the optimization stuck with floaters close to input cameras and unjustified increase in the Gaussian density, 3DGS Kerbl et al. (2023) reset the opacity of all Gaussians primitives with an opacity value greater than 0.01 to 0.01, and prune those with opacities below this threshold. However, numerous Gaussians exhibit minimal visibility contribution and add little to rendered outputs, a fixed small threshold is suboptimal for pruning unnecessary Gaussians during optimization.

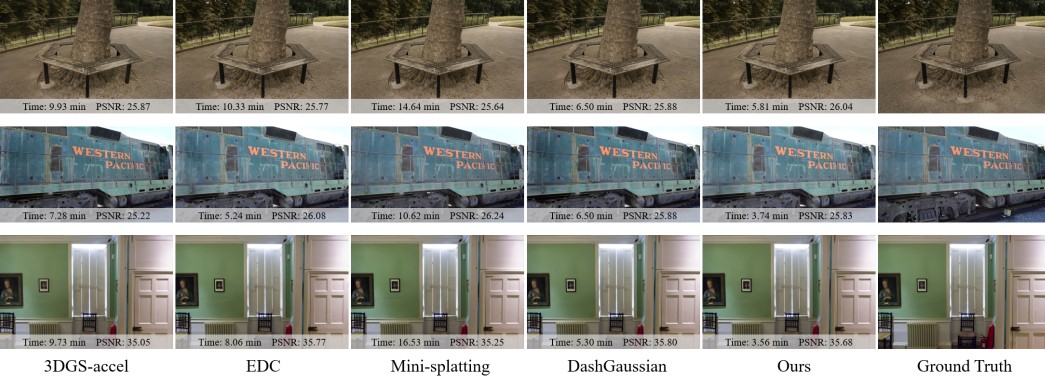

Figure 4: Qualitative comparison between our method and prior 3DGS approaches, along with the corresponding ground truth images from test viewpoints.

To maintain an efficient and compact Gaussian distribution during optimization, we implement an adaptive opacity threshold with a upper limit to prune these visually insignificant and redundant primitives. Let $\boldsymbol{\alpha} \in \mathbb{R}^N$ denote the opacity vector of all $N$ Gaussian primitives. We first sort $\boldsymbol{\alpha}$ in ascending order to obtain $\boldsymbol{\alpha}_{\text{sorted}}$, where the $k$-th element satisfies:

$$\tau_k = \boldsymbol{\alpha}_{\text{sorted}}[k], \quad k = \lfloor N \cdot p \rfloor \tag{7}$$

Here, $p \in (0, 1)$ controls the pruning ratio. The adaptive opacity threshold $\tau$ is then determined with a upper limit $\tau_u$ as:

$$\tau = \min(\tau_k, \tau_u) \tag{8}$$

This pruning operation with the dual-constrained threshold effectively eliminates redundant Gaussians while preserving structurally important primitives.

Table 2: Quantitative evaluation comparing the proposed method with existing 3DGS optimization works. We report SSIM, PSNR (dB), LPIPS, number of Gaussian Primitives and training time (mins). The proposed method achieves superior performance with much less time cost.

| Method | MipNeRF-360 Barron et al. (2022) | | | | | Deep Blending Hedman et al. (2018) | | | | | Tanks & Temples Knapitsch et al. (2017) | | | | |
|---|---|---|---|---|---|---|---|---|---|---|---|---|---|---|---|
| | SSIM ↑ | PSNR ↑ | LPIPS ↓ | $N_{GS}$ ↓ | Time ↑ | SSIM ↑ | PSNR ↑ | LPIPS ↓ | $N_{GS}$ ↓ | Time ↑ | SSIM ↑ | PSNR ↑ | LPIPS ↓ | $N_{GS}$ ↓ | Time ↑ |
| 3DGS Kerbl et al. (2023) | 0.8263 | 27.72 | 0.2016 | 2.578 M | 25.01 | 0.9075 | 29.44 | **0.2381** | 2.475 M | 23.32 | 0.8471 | 23.62 | 0.1772 | 1.576 M | 15.80 |
| Mini-splatting Fang & Wang (2024a) | 0.8325 | 27.56 | 0.2011 | **0.493 M** | 18.21 | 0.9085 | 30.01 | 0.2409 | **0.555 M** | 15.51 | 0.8467 | 23.45 | 0.1804 | **0.301 M** | 10.54 |
| 3DGS-accel Mallick et al. (2024) | 0.8213 | 27.57 | 0.2095 | 2.331M | 11.18 | 0.9027 | 29.54 | 0.2537 | 2.394M | 8.16 | 0.8460 | 23.58 | **0.1756** | 1.550M | 7.73 |
| EDC Deng et al. (2024) | **0.8342** | 27.86 | **0.1964** | 1.253M | 10.41 | 0.9093 | 29.92 | 0.2415 | 0.623M | 7.57 | **0.8496** | 23.98 | 0.1771 | 0.570M | 6.68 |
| DashGaussian Chen et al. (2025) | 0.8261 | **27.90** | 0.2084 | 2.081M | 6.34 | 0.9026 | 30.01 | 0.2511 | 1.955M | 5.12 | 0.8468 | 23.95 | 0.1824 | 1.198M | 5.57 |
| Ours | 0.8257 | 27.79 | 0.2136 | 1.469M | **5.33** | **0.9094** | **30.05** | 0.2540 | 1.272M | **4.54** | 0.8461 | **24.06** | 0.1891 | 0.867M | **4.10** |

## 5 EXPERIMENTS

**Datasets and metrics.** We perform experiments on three real-world datasets: MipNeRF-360 Barron et al. (2022), Deep Blending Hedman et al. (2018) and Tanks&Temples Knapitsch et al. (2017). Following the default data pre-processing in the 3D Gaussian Splatting Kerbl et al. (2023), we initialize the Gaussian primitives with the point clouds extracted from the structure from motion (SFM). we selected one out of every eight images to evaluate the average peak signal-to-noise ratio (PSNR), structural similarity index (SSIM) Wang et al. (2004) and learned perceptual image patch similarity (LPIPS) Zhang et al. (2018). Additionally, we report the number of Gaussian primitives and average training time (in minutes) on each dataset to prove the efficiency of the proposed method.

**Implementation details** We build our method upon the open-source accelerated version of 3DGS code base. Following Kerbl et al. (2023), we train our models for 30K iterations across all scenes. We extend the iteration of densification $\mathbf{T}_{\text{densify}}$ to 25K and set the default max scale factor $K$, pruning ratio $p$, and pruning upper limit $\tau_u$ to 8, 0.03 and 0.05, respectively. To encourage efficient spatial diffusion of Gaussian primitives, we keep the positional learning rate constant during training with downsampled resolutions and reduce it after restoring full resolution. All experiments are conducted on an NVIDIA GeForce RTX 3090 GPU with a AMD EPYC 7413 24-Core processor CPU to ensure a fair comparison.

## 5.1 QUANTITATIVE RESULTS

As shown in Table 2, We report the comparison with the state-of-the-art (SOTA) 3DGS reconstruction methods, i.e., 3DGS Kerbl et al. (2023), 3DGS-accel[1], EDC Deng et al. (2024), mini-splatting Fang & Wang (2024a), and DashGaussian Chen et al. (2025) in Table 2 in terms of training time, the number of Gaussian primitives and standard visual quality metrics. It is worth noting that mini-splatting is built upon 3DGS, whereas EDC and DashGaussian are based on 3DGS-accel. As DashGaussian is not publicly available, we re-implement it based on the methodology described in the paper to serve as the state-of-the-art baseline for comparison.

Compared to the 3DGS-accel, our approach demonstrates a significant $2\times$ speedup with $40\%$ fewer Gaussian primitives across all three datasets. Thanks to the proposed efficient global-to-local optimization and energy-aware multi-resolution densification strategies , our method not only improves computational efficiency but also enhances reconstruction quality. Specifically, it achieves an average improvement of +0.004 in SSIM and +0.31 dB in PSNR , while maintaining strong perceptual fidelity with only a negligible 0.0049 increase in LPIPS. In comparison to existing SOTA methods, our approach achieves the fastest convergence speed while preserving competitive rendering quality.

MSv2 Fang & Wang (2024b) is an extended version of mini-splatting Fang & Wang (2024a), which adopts an aggressive densification strategy and limits the optimization of Gaussian primitives to 18K iterations. For a fair comparison, we also train our proposed method

Table 3: Comparison to Msv2 within 18K optimization iterations.

| Method | SSIM ↑ | PSNR ↑ | LPIPS ↓ | $N_{GS}$↓ | Time ↑ |
|---|---|---|---|---|---|
| MSv2 Fang & Wang (2024b) | 0.8206 | 27.35 | 0.2149 | **0.618 M** | 3.55 |
| Ours-18K | **0.8237** | **27.65** | **0.2137** | 1.085 M | **3.47** |

for 18K iteration, with 15K iterations allocated to densification. Results on the MipNeRF-360 dataset, as shown in Table 3, demonstrates that our method achieves a better performance with a less training time.

## 5.2 QUALITATIVE RESULTS

The qualitative performance is displayed as rendered images in Fig. 4. These results align well with the quantitative results provided in Table 2.

Our method achieves comparable or even better rendering quality with a less training time. Besides, due to the efficient diffusion of the Gaussian primitives in space, our method enables accurate reconstruction of small objects (i.e., lamp bulbs) and produces clear renderings for distant views (i.e., remote house), shown in Fig. 5. Although the limited projected 2D pixel coverage of small and distant objects prevents the improvement from being clearly reflected in the quantitative metrics, the visual results highlight practical benefits that go beyond numerical measurements. These findings underscore the effectiveness and real-world applications of the proposed method.

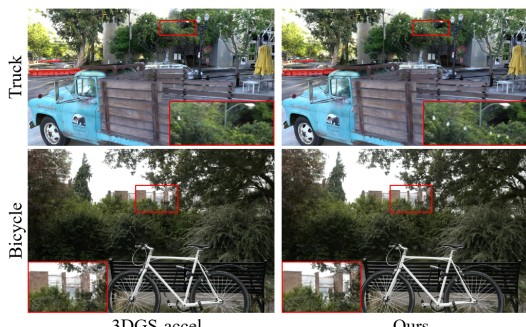

Figure 5: Qualitative results for small and distant object reconstruction.

## 5.3 ABLATION STUDIES

We use the 3DGS-accel Mallick et al. (2024) as our backbone framework and individually integrate each densification method to systematically explore their respective effects on rendering quality and optimization speed. Our experiments are conducted on the MipNeRF-360 dataset Barron et al. (2022), as it comprehensively includes both indoor and outdoor scenes.

---

[0]3DGS-accel denotes the application of efficient per-splat backpropagation and sparse Adam optimizer from taming 3DGS Mallick et al. (2024) to 3DGS Kerbl et al. (2023).

**Global-to-local densification**  We adopt the same configurations as the 3DGS-accel and use iteration $\mathbf{T}_2$ (refer to eqn. 6) as the boundary between the global spread and local refinement.

**Coarse-to-fine densification**  Following the Sec. 4.3, we calculate the number of iterations for different resolution of each scene. The impact is evaluated with and without global-to-fine strategy.

As shown in Tab. 4, the global-to-local strategy reduces computational cost but slightly degrades image quality. In contrast, the coarse-to-fine densification improves PSNR and maintains low LPIPS, while also reducing computational overhead. Combining both global-to-local and coarse-to-fine components further optimizes efficiency without significant loss in quality. We also test the effect of adaptive opacity pruning by applying it to 3DGS-accel. It can reduces limited computational cost but introduces a minor trade-off in image quality. Ultimately, the full model achieves the lowest optimization time with superior performance than baseline. These findings underscore the efficacy of our method in enhancing both the visual fidelity and computational efficiency.

Table 4: Ablation studies of the proposed method on the MipNeRF-360. G2L and C2F denote global-to-local and coarse-to-fine densification.

| Method | SSIM ↑ | PSNR ↑ | LPIPS ↓ | $N_{GS}$↓ | Time ↑ |
|---|---|---|---|---|---|
| 3DGS-accel | 0.8213 | 27.57 | **0.2095** | 2.331 M | 11.18 |
| + G2L | 0.8066 | 27.47 | 0.2235 | 1.887 M | 8.46 |
| + C2F | 0.8246 | **27.84** | 0.2202 | 2.018 M | 7.56 |
| + G2L + C2F | 0.8176 | 27.75 | 0.2203 | 1.853 M | 6.95 |
| +Pruning | 0.8211 | 27.56 | 0.2234 | 1.685 M | 9.21 |
| Full | **0.8257** | 27.79 | 0.2136 | **1.469M** | **5.33** |

**Hyperparameters**  We evaluate the impact of various hyperparameters on actual training efficiency and final reconstruction quality, including densification iteration $\mathbf{T}_{\text{densify}}$ (25K) in Tab. 5, pruning ratio $p$ (0.03) and pruning upper limit $\tau_u$ (0.05) in Tab. 6. The numbers in parentheses denote the default values.

A smaller $\mathbf{T}_{\text{densify}}$ indicates that densification is completed earlier, leaving more iterations for full-precision optimization. However, this typically leads to increased computational cost with only marginal performance improvement. For opacity pruning, a lower pruning ratio $p$ and a lower pruning upper limit $\tau_u$ preserve more Gaussian primitives with small opacity values, which in turn increases computational overhead. In contrast, an aggressive pruning may lead to excessive removal of informative primitives, leading to a noticeable decline in reconstruction quality. Overall, there exists a trade-off between training efficiency and rendering fidelity.

Table 5: Ablation studies of the densification iteration $\mathbf{T}_{\text{densify}}$.

| $\mathbf{T}_{\text{densify}}$ | SSIM ↑ | PSNR ↑ | LPIPS ↓ | $N_{GS}$↓ | Time ↑ |
|---|---|---|---|---|---|
| 15 K | **0.8272** | **27.81** | **0.2115** | 1.476 M | 6.14 |
| 20 K | 0.8268 | 27.79 | 0.2141 | **1.426 M** | 5.73 |
| 25K | 0.8257 | 27.79 | 0.2136 | 1.469 M | **5.33** |

Table 6: Ablation studies of the pruning hyperparameters.

| $p$ | $\tau_u$ | SSIM ↑ | PSNR ↑ | LPIPS ↓ | $N_{GS}$↓ | Time ↑ |
|---|---|---|---|---|---|---|
| fixed $\tau = 0.01$ | | 0.8272 | 27.86 | 0.2098 | 1.682 M | 7.04 |
| 0.01 | 0.05 | 0.8295 | 27.88 | 0.2112 | 1.524 M | 5.74 |
| 0.03 | 0.05 | 0.8257 | 27.79 | 0.2136 | 1.469 M | 5.33 |
| 0.05 | 0.05 | 0.8235 | 27.68 | 0.2168 | 1.412 M | 5.24 |
| 0.03 | 0.01 | 0.8291 | 28.04 | 0.2081 | 1.812 M | 6.84 |
| 0.03 | 0.05 | 0.8257 | 27.79 | 0.2136 | 1.469 M | 5.33 |
| 0.03 | 0.10 | 0.8185 | 27.59 | 0.2277 | 1.286 M | 4.80 |

## 6  CONCLUSION AND LIMITATIONS

In this paper, we present a simple but efficient approach to accelerate 3D Gaussian Splatting for efficient 3D scene reconstruction by decomposing the densification. Through systematic analysis, we reveal that split operations primarily govern global spatial spread of Gaussian primitives, while clone operations focus on local refinement. Leveraging this insight, we propose a global-to-local densification strategy that decouples split and clone operations across training phases, enabling efficient scene coverage followed by detail-preserving refinement. Subsequenctly, we introduce an energy-guided coarse-to-fine multi-resolution framework and a dynamic pruning mechanism to further enhance acceleration. Numerous experiments across three real-world datasets highlight the effectiveness of our strategy in balancing computational efficiency with high-fidelity rendering. This paper aim at a training acceleration and does not address the inherent blur issue in 3DGS, which stems from insufficient gradient accumulation of big Gaussians. We will consider how to design a reasonable gradient threshold to achieve better renderings.

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
