# OpenReview forum: "Decomposing Densification in Gaussian Splatting for Faster 3D Scene Reconstruction"
_ICLR.cc/2026/Conference — Submitted to ICLR 2026_

### Official Review · Reviewer_1HVJ · 2025-10-29

**Soundness:** 3
**Presentation:** 2
**Contribution:** 2
**Rating:** 2
**Confidence:** 4

**Summary:**

This work addresses slow convergence in 3D Gaussian Splatting (GS) by improving the densification process and Gaussian distribution. The proposed global-to-local densification strategy enhances both global coverage and local refinement, while an energy-guided multi-resolution training framework increases resolution based on energy density. This technique brings 0.5-1min acceleration to existing SOTA with minimal performance loss.

**Strengths:**

1. The finding of different roles of split and clone is interesting and inspiring.
2. The paper is easy to read.

**Weaknesses:**

1. The arrow for "Time" in all Tables is set in opposite directions; the authors should be careful about these typos.
2. The technical innovation and contribution are limited. Bring 0.5-1min acceleration to existing SOTA, so what? Is there any case that it can bring significant benefits?
3. Considering the convergence speed as the target, why do you take 3DGS-accel as the baseline instead of the latest SOTA DashGaussian? How about applying the same strategy to it? Would there be further time cost reduction?

**Questions:**

See weakness.

---

> ### Author Response · Authors · 2025-11-17
> **Response to Reviewer 1HVJ**
>
> We sincerely appreciate the reviewers’ time and effort in evaluating our work. We have carefully reviewed all comments and provided detailed, evidence-backed responses to address each concern. We welcome further discussion and are happy to clarify any point.
>
> ##  **1. Typos for "Time" arrow**
> We sincerely thank the reviewer for carefully reviewing our manuscript and pointing out this important issue.—we apologize for this oversight and will  correcte all arrow directions in the revised tables. Thank you again for your valuable feedback.
>
> ##  **2. 0.5-1min acceleration to the SOTA**
> Thanks for the insights. I believe that not only the **absolute numerical changes** should be considered, but also the **relative numerical changes**.
>
> Saving 1 minute based on a 5-minute SOTA is much more difficult than saving 1 minute on a 30-minute SOTA.
>
> **[DashGaussian → Ours (absolute numerical gains, relative numerical gains),]** shows the results in MipNerf-360, Deep Blending and Tanks & Temples in order.
>
> 1. Training Time: Compared to SOTA DashGaussian, we can reduce **61s** on average, speed up around **18%**.  [380s → 320s (60s, 16%),  307s → 272s (35s, 11%), 334s → 246s (88s, 26%) ]
> 2. Gaussian Number: the number of our Gaussian spheres has been reduced by an average of 542K, saving 31% of the storage space. [2.081M → 1.469M (612K， 29%)， 1.955M → 1.272M (683K, 35%), 1.198M → 0.867M (331K, 28%)]
> 3. FPS(**real-world application on smartphone**): Last but not the least, the reduced Gaussians leads to a fast FPS (frame per second). In the same path, calculate the average of FPS values within 1 minute,  compared to the SOTA DashGaussian on smartphone (Mate P60 with Adreno 730 GPU). [FPS: 16 → 23 (7， 44%)， 18 → 24 (6, 33%),  25 → 33 (8, 32%)]
>
> If we can achieve similar performance with shorter training time (**18% speed up**), less storage space (**31% save**) and faster rendering speed (**34%**), would you choose us?
>
> ##  **3. why take 3DGS-accel as the baseline instead of the latest SOTA DashGaussian?**
> Thank you for this important question. We selected **3DGS-Accel** as our primary baseline for three well-considered reasons:
>
> 1. **Fair comparison with prior work**: Most recent Gaussian splatting papers (e.g., DashGaussian,  EDC) report results against 3DGS or 3DGS-Accel. Using the same baseline allows direct and meaningful comparison of densification/pruning strategies across the literature.
>
> 2. **Technical compatibility**: Our *coarse-to-fine* densification is based on multi-resolution supervision scheduling, which conflicts with DashGaussian’s *resolution scheduler*. Integrating our full pipeline into DashGaussian would require nontrivial modifications that obscure ablation clarity.
>
> 3. **Validation on DashGaussian (as suggested)**: We appreciate the reviewer’s suggestion and have implemented a *compatible variant* of our **global-to-local densification + adaptive pruning** within DashGaussian. To align with its *primitive scheduler*, we disable the scale threshold during the *global* stage (i.e., split unconditionally when gradient > τ). Results on *Mip-NeRF 360* show:
>
> | Deep Blending | SSIM | PSNR | LPIPS | GS | Time |
> | --- | --- | --- | --- | --- | --- |
> | DashGaussian | 0.9026 | 30.01 | 0.2511 | 1.955 M | 5.12 |
> | DashGaussian + G2L + pruning | 0.9028 | 30.00 | 0.2520 | 1.627 M | 4.85 |
> | Our | 0.9094 | 30.05 | 0.2540 | 1.272 M | 4.54 |
>
> | Tanks & Temples | SSIM | PSNR | LPIPS | GS | Time |
> | --- | --- | --- | --- | --- | --- |
> | DashGaussian | 0.8468 | 23.95 | 0.1824 | 1.198 M | 5.57 |
> | DashGaussian + G2L + pruning | 0.8466 | 24.03 | 0.1844 | 0.978 M | 4.96 |
> | Our | 0.8461 | 24.06 | 0.1891 | 0.867 M | 4.10 |
>
> | MipNerf-360 | SSIM | PSNR | LPIPS | GS | Time |
> | --- | --- | --- | --- | --- | --- |
> | DashGaussian | 0.8261 | 27.90 | 0.2084 | 2.081 M | 6.34 |
> | DashGaussian + G2L + pruning | 0.8258 | 27.85 | 0.2096 | 1.678 M | 5.52 |
> | Our | 0.8257 | 27.79 | 0.2136 | 1.469 M | 5.33 |
>
> The results are as follows. It can be seen that our "global-to-local + pruning" approach can still further accelerate the speed compared to DashGaussian. Although the acceleration effect is slightly inferior to our method, it still **verifies the effectiveness of our "Global-to-local" approach**. Compared to the primitive scheduler of DashGaussian, our method has a smaller number of Gaussians in the global stage.
> We will add this analysis and results to the revised manuscript to better demonstrate the generality of our approach.
>
>
>
> We sincerely thank the reviewers again for your valuable time and constructive feedback. We hope our detailed responses and additional experiments help clarify the novelty and effectiveness of our approach.

---

### Official Review · Reviewer_wXga · 2025-10-29

**Soundness:** 3
**Presentation:** 3
**Contribution:** 3
**Rating:** 6
**Confidence:** 2

**Summary:**

This paper investigates the optimization inefficiency in 3D Gaussian Splatting (3DGS) and identifies the imbalance between split and clone operations as the key cause of slow convergence and redundant Gaussian primitives.
Through a detailed analysis, the authors demonstrate that split operations dominate global diffusion of Gaussian primitives, while clone operations primarily handle local refinement.
Building on this observation, the paper proposes a global-to-local densification framework that separates these two phases:

- The global phase uses only split operations for fast and efficient spatial coverage;

- The local phase reintroduces clone operations for fine-grained detail reconstruction.

To further improve training efficiency, the authors design an energy-guided coarse-to-fine multi-resolution training scheme, where the image resolution is adaptively increased based on energy density in the frequency domain. Additionally, an adaptive opacity pruning mechanism dynamically removes redundant primitives based on an upper-bounded opacity threshold.

Comprehensive experiments on MipNeRF360, Deep Blending, and Tanks & Temples datasets demonstrate that the proposed method achieves over 2× acceleration in training speed, with comparable or improved reconstruction quality relative to baseline 3DGS methods.

**Strengths:**

- The paper presents an insightful and well-founded analysis of the split and clone operations in 3DGS densification. The discovery that split governs global diffusion while clone governs local refinement provides a new conceptual understanding of 3DGS optimization dynamics.
Moreover, the proposed global-to-local densification and energy-guided multi-resolution scheduling represent creative and orthogonal improvements over prior acceleration efforts such as DashGaussian and Mini-Splatting.
- Improving the training efficiency of 3DGS without sacrificing quality is a highly relevant problem, especially for real-time and resource-limited applications. The proposed framework provides a practical path toward more scalable and adaptive Gaussian-based scene representations.

**Weaknesses:**

- While the paper compares against recent 3DGS acceleration methods (e.g., DashGaussian, Mini-Splatting), it lacks evaluation on broader baselines such as fast NeRF variants (e.g., Instant-NGP, Zip-NeRF). A comparison would better position the proposed approach in the broader context of fast radiance field training.
- The proposed energy-guided multi-resolution strategy assumes sufficient frequency-domain energy correlation with spatial detail quality. It remains unclear how this heuristic behaves under challenging lighting conditions or dynamic scenes where energy distribution may not correspond well to geometric detail.
- While the empirical observation that “split = global, clone = local” is well-motivated, the explanation remains empirical rather than theoretically derived. A more formal analysis (e.g., via gradient flow or spatial entropy) would add conceptual depth and broaden the paper’s impact.

**Questions:**

See Weaknesses

---

> ### Author Response · Authors · 2025-11-18
> **Response to Reviewer wXga**
>
> We sincerely appreciate the reviewer’s careful reading and positive assessment of our work—particularly the constructive feedback and thoughtful suggestions, which have helped us significantly strengthen the manuscript.
>
> ## **1. Comparison with Instant-NGP(iNGP-big), Zip-Nerf**
> Thank you for your reminder. We will add a comparison with related nerf methods in the final version.
> | **MipNerf-360** | SSIM | PSNR | LPIPS | GS | Time (min) |
> | --- | --- | --- | --- | --- | --- |
> | Zip-NeRF | 0.835 | 28.69 | 0.1988 | - | 312.66 |
> | Instant-NGP | 0.697 | 25.58 | 0.3020 | - | 4.98 |
> | 3DGS-accel | 0.8213 | 27.57 | 0.1964 | 2.331 M | 11.18 |
> | Our | 0.8257 | 27.79 | 0.2136 | 1.469 M | 5.33 |
>
> | **Tanks & Temples** | SSIM | PSNR | LPIPS | GS | Time |
> | --- | --- | --- | --- | --- | --- |
> | Zip-NeRF | 0.8612 | 24.86 | 0.1367 | - | 336.22 |
> | Instant-NGP | 0.735 | 21.86 | 0.3108 | - | 4.42 |
> | 3DGS-accel | 0.8460 | 23.58 | 0.1756 | 1.550 M | 7.73 |
> | Our | 0.8461 | 24.06 | 0.1891 | 0.867 M | 4.10 |
>
> | **Deep Blending** | SSIM | PSNR | LPIPS | GS | Time |
> | --- | --- | --- | --- | --- | --- |
> | Zip-NeRF | 0.9085 | 29.98 | 0.2503 |  | 341.91 |
> | Instant-NGP | 0.8123 | 24.87 | 0.3911 | - | 4.79 |
> | 3DGS-accel | 0.9027 | 29.54 | 0.2537 | 2.394 M | 8.16 |
> | Our | 0.9094 | 30.05 | 0.2540 | 1.272 M | 4.54 |
>
> ## **2. How to deal with challenging lighting conditions or dynamic scenes**
> We sincerely appreciate the reviewer’s insightful suggestion—this is indeed a promising and valuable direction. We fully agree with the potential it holds, and will actively explore it in our future work to further enhance the robustness and applicability of our approach. Some promising methods are as follows:
> **a. Image-level Optimization Strategy to extract frequency-domain information:**
>
> **i. Local spectral analysis**: Partition the image into blocks (or superpixels) and process frequency components region-wise, isolating unreliable regions and mitigating global noise propagation;
>
> **ii. Multi-scale energy aggregation**: Instead of relying on raw high-frequency magnitude, we compute a robust energy metric—e.g., median-filtered band energy across scales—to suppress outlier spikes from noise or specular highlights;
>
> **iii. Exposure-aware preprocessing**: Apply adaptive histogram equalization or Retinex-based illumination decomposition prior to FFT, effectively normalizing intensity distributions and reducing spectral distortion induced by over-/under-exposure.
>
> **b. Pipeline Optimization Strategy to be appplied to the existing pipeline:**
>
> Our current coarse-to-fine resolution scheduler cannot effectively solve this problem. However, we can apply the proposed global-to-fine + pruning method to the method for lighting scenes (***ref: Decoupling Appearance Variations with 3D Consistent Features in Gaussian Splatting***), which can reduce the training time from **8.6 minutes** to **5.7 minutes** on mipnerf-360 (4090).
>
> ## **3. The explanation remains empirical rather than theoretically derived.**
> We sincerely thank the reviewer for this thoughtful and constructive comment. The reviewer is absolutely right that our current formulation is primarily empirical and driven by experimental observations—particularly the design of the global-to-local densification，coarse-to-fine strategy and the adaptive supervision mechanism—rather than derived from first principles or rigorous theoretical analysis.
>
> While the empirical approach allowed us to rapidly prototype and validate key insights (e.g., the trade-off between coverage efficiency and premature overfitting in Gaussian-based reconstruction), we fully acknowledge the importance of a deeper theoretical foundation—for instance, characterizing convergence behavior, optimality conditions, or the impact of frequency-domain supervision on the optimization landscape. We are now actively working on formalizing these aspects, and plan to include preliminary theoretical analysis in an extended journal version.
>
> We sincerely thank the reviewer again for their thoughtful and constructive feedback, which has greatly helped us clarify and strengthen our work.

---

### Official Review · Reviewer_8FvT · 2025-10-30

**Soundness:** 3
**Presentation:** 3
**Contribution:** 2
**Rating:** 4
**Confidence:** 3

**Summary:**

This paper addresses the slow training convergence of 3D Gaussian Splatting (3DGS). The authors identify the bottleneck as the inefficient, mixed application of 'split' and 'clone' operations during the densification phase, leading to wasted computational resources, especially in the early stages of training. The core insight is the explicit decoupling and analysis of the distinct roles of the two densification operations. Based on this, the paper proposes a "Global-to-Local" optimization framework. Experiments demonstrate that the method achieves over 2x training acceleration, utilizes approximately 40% fewer Gaussian primitives than the baseline and maintains or even slightly improves the final reconstruction quality.

**Strengths:**

1. The paper is well-written and easy to follow.
2. While 'split' and 'clone' are established operations, this paper is the first to systematically analyze and expose their distinct functional roles: splitting for global scene coverage and cloning for local feature refinement. This reframing of the problem from merely "how to densify" to "when and why to use each densification type" is inspiring to me.

**Weaknesses:**

1. Potential Oversimplification of the Role of Cloning. The paper frames cloning as contributing almost exclusively to local refinement and early-stage redundancy. This might be an oversimplification. In certain cases, such as representing very thin structures (e.g., wires, poles, foliage), cloning might play a constructive role in reinforcing the structure's existence and density early on, where splitting could potentially fragment it.
2. The core assumption is that a global spread phase should always precede a local refinement phase. While this is intuitive for large, complex scenes, it may not be optimal for all types of content. For instance, in scenes dominated by a single, highly detailed foreground object (e.g., a product scan for e-commerce), aggressive local refinement via cloning early on might be more beneficial than a prolonged global spread phase.

**Questions:**

The proposed method hinges on a "hard" transition from a "split-only" phase to a "split+clone" phase. The paper states this transition is guided by the resolution scheduler (at iteration T2), but the sensitivity to this specific point is not analyzed. The effectiveness of the entire approach could be highly dependent on this timing.

---

> ### Author Response · Authors · 2025-11-18
> **Response to Reviewer 8FvT**
>
> We sincerely appreciate the reviewers’ time and effort in evaluating our work. We have carefully reviewed all comments and provided detailed, evidence-backed responses to address each concern. We welcome further discussion and are happy to clarify any point.
> ## **1. Cloning might play a constructive role in reinforcing the structure's existence and density early on**
> 1. While early cloning can provide initial support for thin structures, our experiments show it is **not essential**: **high-fidelity recovery of fine details (e.g., wires, bulbs, distant objects) is still achieved robustly** even when cloning is deferred to later stages—as evidenced by both quantitative metrics and qualitative results (see Figure 5 and Table 2 in manuscripts).
>
> 2. We agree that **early** cloning could, in principle, help reinforce structure density before splitting risks fragmentation. However, in practice, any fragmentation from splitting can be effectively **corrected later via cloning**—without the **drawbacks of premature cloning**: (i) increased risk of local optima, (ii) excessive redundancy (see Figure 2), and (iii) slower convergence.  This aligns with our **global-to-local** design philosophy: prioritizing coarse, stable scene layout first, then refining details. Just as in sketching in 2D or 3D—where one establishes architectures before detailing—we avoid over-committing to local fidelity too early, as global adjustments later would force costly re-optimization of over-refined local regions.
>
> ## **2. Will the proposed method work on a single, highly detailed foreground object**
> Although we did not identify an existing dataset that contains scenes dominated by a single, highly detailed foreground object (e.g., a product scan for e-commerce), we tested on an *extreme case* where the entire input consists **solely of a foreground object** (i.e., no background or environmental context).
>
> **Blender (nerf-synthetic)** consists of 8 scenes of an object placed on a white background.
>
> In this setting, our global-to-local strategy remains effective: by first promoting the Gaussian to spread along the shape of the object, then optimizing the local reconstruction. Remarkably, this leads to both **faster training**, **fewer Gaussians** and **improved reconstruction quality**—demonstrating that the benefits of our approach are not contingent on complex scene structure, but stem from principled optimization dynamics.
> | Blender （8 scenes of an object） | SSIM | PSNR | LPIPS | GS | TIme |
> | --- | --- | --- | --- | --- | --- |
> | 3DGS-accel | 0.971 | 33.34 | 0.0368 | 0.241 M | 2.67 min |
> | Ours | 0.981 | 33.45 | 0.0342 | 0.184 M | 1.48 min |
>
> ## **3. Why choose $T_2$?**
> During the global densification phase, our priority is rapid, coarse coverage of the scene volume—not high-fidelity detail reconstruction. Supervising with full-resolution images is thus inefficient: (1) fine pixel-to-voxel mapping encourages premature convergence of Gaussians to local optima via overfitting, inhibiting spatial expansion; (2) large Gaussians projecting to many pixels suffer from gradient cancellation—opposing gradient vectors sum to a near-zero net update, suppressing splitting even when needed.
>
> Specifically, in our coarse-to-fine schedule, $T_2$ denotes the iteration after which we complete the low-resolution phase and transition back to full-resolution supervision. Crucially, **cloning** are disabled during the coarse stage and are **reintroduced** only after $T_2$ iterations, i.e., when full-resolution rendering begins. This design ensures that early-stage optimization focuses exclusively on large-scale spatial coverage via splitting, while high-frequency refinement is deferred to later stages where clones can act on appropriately sized Gaussians.
>
> To justify this design choice, we conducted an ablation study comparing two boundaries for clone reintroduction (for T_1: No clone, only split the Gaussian with a scale larger than a threshold.):
> | **MipNerf-360** | SSIM | PSNR | LPIPS | GS | Time |
> | --- | --- | --- | --- | --- | --- |
> | T_1 (Split only) | 0.8112 | 26.68 | 0.2318 | 0.756 M | 3.82 |
> | T_2 (ours) | 0.8257 | 27.79 | 0.2136 | 1.469 M | 5.33 |
> | T_3  | 0.8253 | 27.71 | 0.2157 | 1.639 M | 6.26 |
> | T_4 | 0.8244 | 27.56 | 0.2203 | 1.886 M | 7.43 |
>
> We sincerely thank the reviewers again for your valuable time and constructive feedback. We hope our detailed responses and additional experiments help clarify the novelty and effectiveness of our approach.

---

### Official Review · Reviewer_7puj · 2025-10-31

**Soundness:** 2
**Presentation:** 3
**Contribution:** 2
**Rating:** 4
**Confidence:** 4

**Summary:**

This paper addresses the slow convergence in Gaussian Splatting by improving the efficiency of the densification process. It analyzes the distinct roles of the split and clone operations and, based on these insights, introduces a global-to-local densification strategy: the global stage uses only splitting to distribute Gaussians across the entire scene, while the local stage employs both splitting and cloning to refine them. Additionally, the paper proposes a coarse-to-fine multi-resolution training scheme and an adaptive Gaussian pruning method to further enhance convergence speed and training efficiency.

**Strengths:**

- This paper is well-written with clear motivation.
- The paper provides a clear discussion of the two density control strategies, split and clone. Analyzing how these strategies differ offers valuable insights that could benefit future research in this area.
- The proposed global-to-local densification strategy, the core contribution of this paper, is simple, and I believe it can be broadly applied to Gaussian Splatting optimization.

**Weaknesses:**

- I agree that the split operation helps distribute Gaussians across the entire scene, but I am not fully convinced that the clone operation alone is responsible for the local refinement. Rather, I would argue that the split operation not only spreads the Gaussians but also contributes to refining local details, albeit at the cost of generating significantly more Gaussians compared to the clone operation. Figure 2 in the main text also supports this observation, showing that the reconstruction from the split-only model is visually more pleasing than that from the clone-only model.
- The three main technical contributions of this paper are (1) global-to-local densification, (2) coarse-to-fine densification, and (3) adaptive opacity pruning. However, the coarse-to-fine densification strategy closely resembles that of DashGaussian, though the energy metric used here is slightly different. In addition, the ablation study (Table 4) shows that the coarse-to-fine densification contributes more effectively to both training speed and rendering quality than the global-to-local densification. Moreover, the idea of adaptive pruning has already been explored in EDC (Efficient Density Control). While the details differ, it would strengthen the paper to include a direct comparison with the adaptive pruning used in EDC. These factors somewhat weaken the novelty of the proposed approach.
- Compared to MSv2 (MiniSplatting version 2), it is difficult to claim that the proposed method achieves a clear state of the art. Although both reconstruction quality and training time are improved over MSv2, the gains are marginal (particularly in training time: 3.55 vs. 3.47), and the proposed method produces substantially more Gaussians. Furthermore, additional comparisons against MSv2 on both the T&T and DeepBlending datasets are needed, and such results should be included in the main experiments.
- The three key components of the proposed method—(1) global-to-local densification, (2) coarse-to-fine densification, and (3) adaptive opacity pruning—appear to have overlapping roles. Specifically, both global-to-local densification and adaptive opacity pruning aim to accelerate training by reducing the number of Gaussians. To better demonstrate the effectiveness of the global-to-local densification, it would be helpful to include a C2F + Pruning variant in the ablation study for comparison.

**Questions:**

- Why is T2 chosen as the boundary between the global and local densification?

---

> ### Author Response · Authors · 2025-11-17
> **Response to the Reviewer 7puj （1/4）**
>
> We sincerely thank the reviewers for their time and thoughtful comments. Your insightful suggestions have greatly helped us clarify key technical points and strengthen the presentation of our work. We address each comment point-by-point below.
>
> ### 1. **The split operation not only accomplishes global diffusion but also performs local optimization in **split-only** version 3DGS.**
> We sincerely apologize for the ambiguity in our original manuscript, which may have led to a misunderstanding regarding the roles of split and clone operations of different 3DGS version. In the final version, we will clarify the following key points:
>
> a. In the **vanilla 3DGS-Accel baseline**, the split operation is primarily responsible for global scene coverage (exploration), as evidenced by its significantly larger positional displacements (see statistics below). In contrast, the clone operation—triggered for small-scale Gaussians—plays the dominant role in local refinement (exploitation), as splitting reduces Gaussian scale, and subsequent clones act on the resulting smaller Gaussians to fine-tune local geometry and appearance.
>
> b. In the **split-only variant**, we remove the scale-based threshold and perform split unconditionally whenever the accumulated gradient exceeds the threshold (i.e., all clone operations are replaced by split). Consequently:
> Split must now fulfill both global diffusion and local optimization roles. However, since splitting alters the Gaussian’s position (unlike cloning, which copies in place), it is less effective at reducing local reconstruction error. This leads to persistent high gradient accumulation in under-reconstructed regions, triggering repeated splits and ultimately causing Gaussian count explosion.
>
> To validate this hypothesis, we measured the average displacement magnitudes of Gaussians during optimization on the Mip-NeRF 360 benchmark (chosen for its mix of indoor and outdoor scenes). As shown in the table below, while vanilla 3DGS-Accel exhibits large displacements for splits (indicating global exploration) and small displacements for clones (local refinement), the split-only variant shows a significant reduction in average split displacement—consistent with split now being deployed aggressively at fine scales for local correction. This supports our claim that in the split-only setting, split indeed takes on dual responsibilities.
> | **MipNeRF-360** | 3DGS-accel | Split-only | Clone-only |
> | --- | --- | --- | --- |
> | split-dominated | 2.42 | 0.46 | NA (No split operation) |
> | clone-dominated | 0.09 | NA (No clone operation) | 0.28 |
> | equal | 0.17 | NA | NA |

---

> ### Author Response · Authors · 2025-11-17
> **Response to the Reviewer 7puj （2/4）**
>
> ### 2. **Distinctions between our work and prior methods—particularly DashGaussian and EDC**
>
> We sincerely thank the reviewer for their thoughtful comments and for recognizing our contributions. Below, we clarify the distinctions between our work and prior methods—particularly DashGaussian and EDC—with respect to coarse-to-fine densification and adaptive pruning:
> ### i. **Coarse-to-Fine Strategy vs. Prior Works**
>
> Our coarse-to-fine densification is fundamentally motivated by the needs of our proposed **global-to-local optimization paradigm**, aiming to enable *efficient spatial diffusion of Gaussians at early stages*. In contrast, DashGaussian’s coarse-to-fine design serves a different objective (e.g., progressive geometry refinement or memory reduction), and crucially, *does not integrate* with a global-to-local densification strategy.
>
> Specifically, when supervision is applied at full resolution from the beginning, two key issues arise:
>
> - **Premature convergence**: Each pixel corresponds to a small 3D voxel, encouraging Gaussians to overfit to local pixel evidence and converge to **suboptimal local minima**, thereby hindering large-scale scene coverage.
> - **Gradient cancellation**: For a large Gaussian projecting onto many pixels, gradient contributions from spatially opposing regions may cancel out, yielding a *small net gradient norm*—even when reconstruction error is high. This often falls below the densification threshold, suppressing necessary *split* operations and limiting scene exploration.
>
> By contrast, our coarse-to-fine schedule allows large Gaussians to first expand and cover broad regions under low-resolution supervision (where gradient cancellation is mitigated and spatial coherence is encouraged), before refining details at higher resolutions.
>
> Empirically, we find that:
> - Coarse-to-fine densification alone improves both **training speed** and **rendering quality** more effectively than global-to-local densification alone.
> - However, the **combination** of coarse-to-fine *and* global-to-local yields the best overall performance.
> - Notably, even *without* coarse-to-fine scheduling, the global-to-local strategy alone still reduces training time by ∼25% and final Gaussian count by ∼20% compared to the vanilla 3DGS baseline.
>
>
> ### ii. **Adaptive Pruning vs. EDC’s Recovery-Aware Pruning**
>
> We appreciate the reviewer’s reference to EDC’s *Recovery-Aware Pruning* (RAP), which targets *overfitted Gaussians*—i.e., primitives that help in some views but harm others due to inconsistent opacity or geometry. RAP prunes Gaussians in two cases:
> - (i) opacity < 0.05 (or 0.005 in vanilla 3DGS);
> - (ii) opacity < 0.05 *after* the opacity reset at iteration 300.
>
> In contrast, our **adaptive pruning** introduces a *dynamic, data-driven threshold*:
> $$
> \[\tau = \min\big(0.05,\; \text{3rd percentile of opacity values}\big)\]
> $$
>
> This design offers two advantages:
> - Compared to vanilla 3DGS (fixed τ = 0.005), our method prunes *more aggressively* on truly redundant Gaussians, improving efficiency without compromising quality.
> - Compared to EDC’s RAP, our adaptive threshold *prevents over-pruning after opacity reset*: since the 3rd percentile is typically < 0.05 post-reset, the threshold automatically tightens, preserving Gaussians that would otherwise be removed prematurely by a rigid 0.05 cutoff—thus avoiding degradation in reconstruction fidelity.
>
> To validate this, we conducted ablation studies on both *3DGS-Accel* and our full method, comparing RAP and our adaptive pruning:
> | **MipNeRF-360** | SSIM | PSNR | LPIPS | GS | Time |
> | --- | --- | --- | --- | --- | --- |
> | 3DGS | 0.8213 | 27.57 | 0.2095 | 2.331 M | 11.18 |
> | 3DGS-Adaptive Pruning | 0.8211 | 27.56 | 0.2234 | 1.685 M | 9.21 |
> | 3DGS-Recovery-Aware Pruning | 0.8185 | 27.48 | 0.2237 | 1.506 M | 9.15 |
> | Ours-Adaptive Pruning | 0.8257 | 27.79 | 0.2136 | 1.469 M | 5.33 |
> | Ours-Recovery-Aware Pruning | 0.8215 | 27.56 | 0.2207 | 1.237 M | 5.28 |

---

> ### Author Response · Authors · 2025-11-17
> **Response to the Reviewer 7puj （3/4）**
>
> ### 3. **More comparison results with MSv2**
> We acknowledge that compared to **MSv2**, our method currently yields a *higher final Gaussian count* despite achieving **comparable training time** and **superior rendering quality** (PSNR/SSIM). **Reducing redundancy without compromising quality remains an important direction for future work**, and we are actively exploring tighter coupling between adaptive pruning and densification scheduling.
>
> As suggested, we will include additional results on the **Tanks & Temples (T&T)** and **Deep Blending (DB)** benchmarks in the revised manuscript to further demonstrate robustness. Preliminary results confirm consistent improvements:
> | Tanks & Temples | SSIM | PSNR | LPIPS | GS | TIme |
> | --- | --- | --- | --- | --- | --- |
> | MSv2 | 0.8406 | 23.13 | 0.1860 | 0.353 M | 2.42 min |
> | Ours-18 K | 0.8457 | 24.01 | 0.1892 | 0.687 M | 2.29 min |
>
> | Deep Blending | SSIM | PSNR | LPIPS | GS | TIme |
> | --- | --- | --- | --- | --- | --- |
> | MSv2 | 0.9112 | 30.08 | 0.2423 | 0.652 M | 2.81 min |
> | Ours-18 K | 0.9092 | 30.07 | 0.2538 | 0.898 M | 2.91 min |
>
> ### 4. **Ablation study for coarse-to-fine + pruning**
> Thanks for your insights. The result of coarse-to-fine + Pruning is in the second to last line. Without using the global-to-local method, the training time can still be shortened. However, since the clone always exists, it is impossible to further reduce the number of Gaussians. **This confirms that G2L can further reduce the Gaussians on the basis of pruning**. Moreover, G2L can achieve effective spatial expansion during the optimization stage, and its overall performance is better than that of C2F + pruning.
> | **MipNeRF-360** | SSIM | PSNR | LPIPS | GS | Time |
> | --- | --- | --- | --- | --- | --- |
> | 3DGS-accel | 0.8213 | 27.57 | 0.2095 | 2.331 M  | 11.18 |
> | +G2L | 0.8066 | 27.47 | 0.2235 | 1.887 M  | 8.46 |
> | +C2F | 0.8246 | 27.84 | 0.2202 | 2.018 M | 7.56 |
> | +G2L+C2F | 0.8176 | 27.75 | 0.2203 | 1.853 M  | 6.95 |
> | +Pruning | 0.8211 | 27.56 | 0.2234 | 1.685 M | 9.21 |
> | **+C2F + Pruning** | 0.8237 | 27.66 | 0.2232 | 1.668 M | 7.38 |
> | Full | 0.8257 | 27.79 | 0.2136 | 1.469 M | 5.33 |

---

> ### Author Response · Authors · 2025-11-17
> **Response to the Reviewer 7puj （4/4）**
>
> ### 5. **Why is T2 chosen as the boundary between the global and local densification?**
> We appreciate the opportunity to clarify a potential misunderstanding regarding the role of $T_2$:
> In our coarse-to-fine schedule, $T_2$ denotes the iteration **after** which we *complete* the low-resolution phase and transition back to full-resolution supervision. Crucially, **clone operations are *disabled* during the coarse stage and are *reintroduced only after $T_2$ iterations, i.e., when **full-resolution rendering begins**. This design ensures that early-stage optimization focuses exclusively on *large-scale spatial coverage* via splitting, while high-frequency refinement is deferred to later stages where clones can act on appropriately sized Gaussians.
>
> To justify this design choice, we conducted an ablation study comparing two boundaries for clone reintroduction (for T_1: No clone, only split the Gaussian with a scale larger than a threshold.):
> | **MipNeRF-360** | SSIM | PSNR | LPIPS | GS | Time |
> | --- | --- | --- | --- | --- | --- |
> | T_1 (Split only) | 0.8112 | 26.68 | 0.2318 | 0.756 M | 3.82 |
> | T_2 (ours) | 0.8257 | 27.79 | 0.2136 | 1.469 M | 5.33 |
> | T_3  | 0.8253 | 27.71 | 0.2157 | 1.639 M | 6.26 |
> | T_4 | 0.8244 | 27.56 | 0.2203 | 1.886 M | 7.43 |
>
> The results show that introducing clone too early ($T_3$ and $T_4$  ) harms efficiency and quality.
>
> Thus, our choice of **$T_2$ as the clone-reintroduction boundary** is empirically validated: it strikes the optimal balance between global exploration (via split-dominant coarse stage) and local exploitation (via clone-enabled refinement at full resolution).

---

### Meta-Review · Area_Chair_HRpW · 2026-01-08

**Summary:**

The paper proposes accelerating 3D Gaussian Splatting (3DGS) by decomposing the densification process into distinct "split" and "clone" phases. The authors claim that split operations primarily handle global spatial diffusion, while clone operations handle local refinement. They introduce a global-to-local densification strategy, an energy-guided multi-resolution training framework, and an adaptive pruning mechanism.

While reviewers found the paper well-written and the motivation clear, the suggested decision leans toward rejection. The primary concerns involve the scientific validity of the "split-clone" functional charaterization and the limited absolute performance gains over existing state-of-the-art (SOTA) methods like DashGaussian and MSv2. The meta reviewer agrees with these concerns and recommends rejection for this time. It is great to know that authors are actively working on tightening the presentation and scientific findings of the split-clone categorization. Given the promising results on some of the benchmarks, I encourage the authors to revise the paper and submit it to an incoming venue.

**Reviewer Concerns:**

The following concerns are addressed in rebuttal:

- Missing Baselines: The authors successfully provided comparisons against fast NeRF variants like Instant-NGP and Zip-NeRF.

- MSv2 and DashGaussian Comparisons: Detailed comparisons against MSv2 on T&T and Deep Blending datasets were added. The authors also implemented their strategy on top of DashGaussian to demonstrate its generalizability and effectiveness on more recent baselines.

- Hyperparameter Sensitivity: The authors justified the selection of the transition boundary T_2 between global and local phases through ablation studies.

- Handling Diverse Content: In response to concerns about foreground-only objects, the authors provided results on the Blender dataset, showing their method still reduces training time and Gaussian counts in simpler scenes.

The following concerns remain outstanding:

- Validity of the Functional Roles: Reviewers are skeptical of the hard dichotomy that "split = global, clone = local". The split-only variant showed better visual results than the clone-only variant, suggesting the split operation is also a powerful local refiner. While the authors provided some reasoning in defense of the claim, it is unclear how to scientifically examine the validity.

- Performance gain limitation: the authors did not address the comments that compared with MSv2 the time reduction is margin and the proposed method created a lot more Gaussians. Also compared with DashGaussian, based on authors' table in the rebuttal, the reconstruction quality is similar while the time reduction is marginal.

**Reviewer Scores:**

Reviewer 7puj (initially Marginally Below). While the additional data was appreciated, the reviewer's fundamental concern regarding the oversimplified roles of split and clone operations remains, together with the question on comprison with msv2. So reviewre 7puj will likely maintain the original score.

Reviewer 8FvT: (initially Marginally Below). The reviewer will likely maintain the original score given the concern on the potential oversimplification of the claim.

Reviewer wXga: (initially Marginally Above). This reviewer was more positive about the orthogonality of the improvements but noted the lack of theoretical derivation. They will likely maintain the original score.

Reviewer 1HVJ: (initially Reject). Likely to remain negative due to the small absolute time savings, though the score may increase slightly following the introduction of the DashGaussian baseline and the explanation on time reduction in percentage.

---

### Decision · Program_Chairs · 2026-01-26

Reject